# Text Mining in Education—A Bibliometrics-Based Systematic Review

Alireza Ahadi [1,2,*], Abhay Singh [3,*], Matt Bower [2] and Michael Garrett [4]

1   Connected Intelligence Centre, University of Technology Sydney, Sydney 2007, Australia
2   Faculty of Arts, School of Education, Macquarie University, Sydney 2109, Australia; matt.bower@mq.edu.au
3   Department of Applied Finance, Macquarie Business School, Macquarie University, Sydney 2109, Australia
4   Cinglevue International Pty Ltd., Unit 22/1 Walsh Loop, Joondalup 6027, Australia; dr.michael.j.garrett@gmail.com
*   Correspondence: alireza.ahadi@uts.edu.au (A.A.); abhay.singh@mq.edu.au (A.S.)

**Abstract:** Advances in Information Technology (IT) and computer science have without a doubt had a significant impact on our daily lives. The past few decades have witnessed the advancement of IT enabled processes in generating actionable insights in various fields, encouraging research based applications of modern Data Science methods. Among many other fields, education research has also been adopting different analytical approaches to advance the state of education systems. Moreover, developments in software engineering and web-based applications have made collection of education data possible at large scales. This systematic review aims to explore the 21st century's state of the art applications of text mining methods used in the field of education. We analyse the metadata of all publications that use text mining or natural language processing in educational settings to report on the key themes of application of text mining methods in educational studies providing an overview of the current state of the art and the future directions for research and applications.

**Keywords:** text mining; natural language processing; learning analytics; systematic review; bibliometrics; education; teaching and learning

## 1. Introduction

The 21st century is without a doubt significantly impacted by technology. Advances in technology has not only influenced different aspects of human lives by advancing economies and infrastructure, but also contributed to the advancement in the delivery of education and the learning process. Prior to 1990, submission of homework, assignments, and students work was carried out in the traditional pen and paper fashion. Thanks to the advances in technology, the emergence of early **L**earning **M**anagement **S**ystems (LMS) such as FirstClass and EKKO made the electronic submission of students work possible. Nowadays, majority of university, colleges and even schools use such systems which have made systematic and automated collection and exploration of such data much easier. A large proportion of such data is in textual format with a great potential for analytics for educational, research and even industrial purposes. Among different analytical approaches, **N**atural **L**anguage **P**rocessing (NLP) [1] and Text Mining [2] are two of the contemporary areas that have attracted academics, researchers and practitioners in the education community. While the main goal of NLP is to use theoretically motivated range of computational techniques for analysing and representing naturally occurring texts at one or more levels of linguistic analysis, Text Mining is focused more on the processes that derive high-quality information from text. Text mining and NLP have several techniques which can be used to analyse the text generated by educational processes. Considering the relatively recent applications of text mining in the field of education, researchers and practitioners in the education domain may want to investigate some applications of text mining to identify the techniques and algorithms that can be used by education research

community. Systematically reviewing the applications of text mining in education over the past two decades can be helpful in identifying such algorithms and methods.

Systematic reviews [3] are specific type of reviews that use systematic and reproducible methods to identify, select and critically appraise all relevant research related to a particular topic, and aim to collect and analyse data from the studies that are included in the review. More specifically, systematic reviews aim to present general knowledge about a topic and attempt to show the history of the development of knowledge about the topic (see [4] for an example). Multiple systematic reviews have successfully attempted to provide a big picture view of the application of data mining for mathematics and science education [5], educational text mining [6], and application of natural language processing in education [7–9]. In this systematic review paper, we aim to advance the current knowledge of the application of text mining and natural language processing in educational contexts in a general sense, with a focus on the empirical applications of such techniques in teaching and learning. In particular, in this paper, we systematically review the literature from January-2000 to January-2022 to answer the following research questions:

- What has been the state of the art in application of text mining methods in the field of education?
- What are the main themes in using Text mining in education in the 21st century and how have they evolved?

The review found that certain research areas related to the application of text mining and natural language processing are fully developed and have attracted the attention of the research community to an acceptable degree. Examples of such areas include learning analytics, analysis of the MOOC data and writing analytics. Other text mining techniques such as ontology based methods, clustering and machine learning based approaches are not fully utilised. Additional to the insights obtained from the systematic review, the data collection process in this study provides an innovative methodology to search for relevant keywords for a research area of interest for systematic reviews.

The paper is organised as follows. Section 1 provides a brief introduction, explores the research questions investigated by this systematic review and highlights the main findings of the paper. Section 2 is focused on the methodology used for data collection and conducted analysis. Section 3 discusses the main results of the systematic review. Section 4 is dedicated to discussion, aims to highlight strengths and limitations of the work, and draws conclusions in light of the findings of the paper.

## 2. Methodology

*Selection Criteria and Data Collection*

To ensure a systematic review process, the guideline provided by Higgins et al. [10] was used. The systematic literature review process used in this study included four primary steps including formulation of the research questions, setting protocol of systematic review, analysis of the literature, and finally data analysis and reporting of the findings. Many educational research papers have been published in 21st century that integrate text mining or natural language processing in their methodology but not all of them are related to our two research questions. The paper selection criteria is targeted to ensure that our analysis is mainly focused on those peer reviewed research papers that represent the application of the aforementioned techniques in student's learning and improved teaching interventions. We aimed to find these peer reviewed research papers published in 21st century that focus primarily on objectives that related to our research questions. Furthermore, this systematic review aims to discover emerging trends in text mining and natural language processing techniques to deliver insights for the researchers for further investigation.

Figure 1 illustrates the process used to identify the papers to include in the systematic review for this study which was guided by PRISMA guideline [11]. In order to collect all the papers related to educational text mining, two abstraction and citation databases including Web of Science (Core Collection) and Scopus were targeted. We selected these two traditionally famous databases [12] because manual inspection of the conference proceedings and journals covered by these two databases revealed that the combinatory use of these two databases gives us the highest degree of coverage of the author keywords that are related to our study. Therefore, the initial search term was set to find those English peer-reviewed publications that are published in 21st century and have "text mining" or "text analytics" or "text analysis" or "writing analytics" or "natural language processing" or "NLP" or "language model" or "computational linguistics" in their title, and also have "teach*" or "learn*" or "student" or "educat*" or "university" or "college" or "institution" or "school" in their title, abstract or keywords. To accomplish that, the following initial search terms were used (We thank the anonymous reviewers for their invaluable comments enabling a broader keyword search):

- Scopus search term: (TITLE ("text mining" OR "text analytics" OR "text analysis" OR "natural language processing" OR "NLP" OR "writing analytics" OR "writing analysis" OR "language model" OR "computational linguistics") AND TITLE-ABS-KEY ("teach*" OR "learn*" OR "educat*" OR "university" OR "college" OR "institution" OR "school" OR "student")) AND PUBYEAR > 1999 AND (EXCLUDE (DOCTYPE, "re")) AND (LIMIT-TO (LANGUAGE, "English"))
- Web of Science search term: (TI = ("text mining" OR "text analytics" OR "text analysis" OR "natural language processing" OR "NLP" OR "writing analytics" OR "writing analysis" OR "language model" OR "computational linguistics")) AND TS = ("teach*" OR "learn*" OR "educat*" OR "university" OR "college" OR "institution" OR "school" OR "student") and Review Articles (Exclude–Document Types) and English (Languages)

Applying the selection criteria on Scopus and Web of Science returned 4433 and 2331 publications respectively. Upon closer inspection of the returned papers, we noted that a considerable number of key papers of the field are not identified by neither the Web of Science nor the Scopus. This is explained by at least two common reasons: first, in some cases there is no explicit mention of the discipline in the title of the papers; instead the authors chose to use a term that represents a broader discipline (for example "learning analytics" as a discipline instead of "text mining") or used the formal name for a direct application of text mining in educational settings (e.g., "automated writing evaluation"); secondly, for some of the publications the authors put the name of the text analysis technique (e.g., "tf-idf") and/or specific technical word (e.g., "recurrent neural network") that were used to analyse the educational text data explicitly in the title of the paper. Therefore, we needed to extend our search term in ways that it caters for those publications that are potentially related to the scope of this study but are not returned by Scopus or Web of Science when the initial search term is used.

To tackle these issues, we extracted all the keywords present in the bib records of the publications that were identified by the first search, and sort them based on their frequency. Next using z-score transformation, we calculated the z-scores of each author keyword (calculated based on the frequency of each author keyword) and using a cut-off value of +1.96 we selected those author keywords (n = 41) which are enriched in the bib records of the result of our initial search. This gave us a pool of author keywords (See Table 1) that are favourable for this study, providing a basis for extending our search term. Exploring the list of abundant author keywords also made us realise that there are some highly enriched author keywords that are not related to our interest (e.g., "electronic health records"). Later, we used these author keywords in the construction of the new search terms to reduce the number of false positives in the results of our new search. Also, this list helped us identify variations in author keywords (e.g., "language model" and "language models") that should be considered when constructing the new search term.

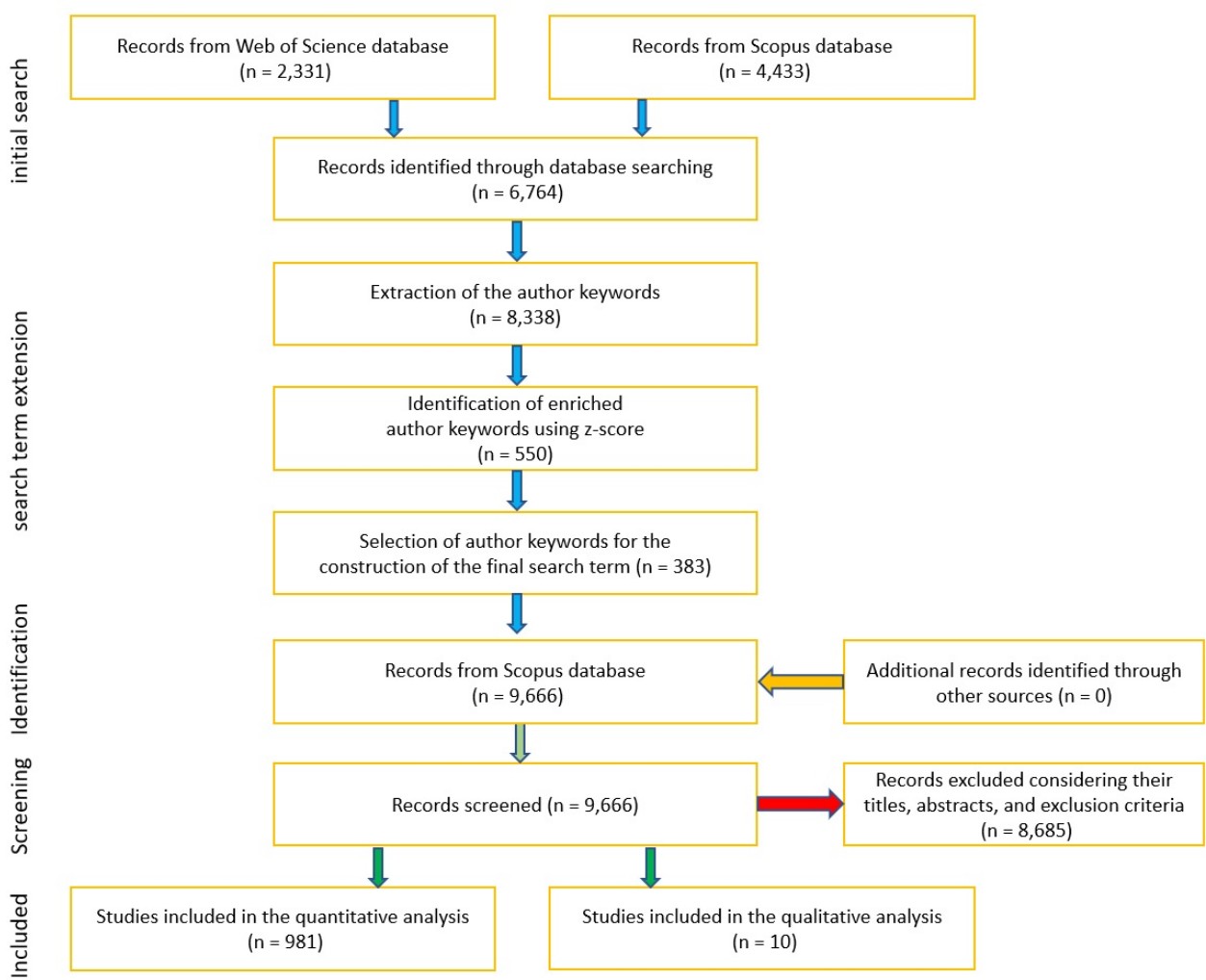

**Figure 1.** PRISMA [11] guided systematic review procedure used for this study.

Next for each item in the prepared list of enriched author keywords, we assigned the author keywords to a group:

- **Education related terms (Group A)**: words that represent education, teaching, or learning (e.g., "distance learning", "MOOCs")
- **Text related jargon (Group B)**: terms that deal with preparing, processing, presenting or analysing text data (e.g., "word embedding", "sentiment analysis")
- **Data analysis technique, jargon or discipline (Group C)**: terms that represent the name of a technique or part of a process that is concerned with the analysis of the data (e.g., "support vector machine", "neural networks")

Categorisation of the 41 author keywords into the aforementioned groups resulted in 1, 20 and 18 author keywords for groups A, B and C respectively. Since we needed more "education" related author keywords, we intuitively relaxed the z-score cut-off so that we can go down deeper in the list to add more author keywords to our defined groups, importantly focusing on keywords related to group A. In the end, we collected 60, 167 and 156 author keywords for groups A, B and C that now can be considered for the construction of the new search terms. Next we define two new groups of search terms and use the author keywords for the implementation of these search terms:

- Publications that have:
  - a text related jargon (Group B) as well as an education related term (Group A) in their title

- Publications that have
  - a data analysis related technique, jargon or discipline (Group C) in their title, and
  - a text related jargon (Group B) in their title, abstract or author keywords and
  - an education related term (Group A) in their title

**Table 1.** Keywords of the papers that were returned when initial search term was used.

| Author Keyword | Group | Frequency |
|---|---|---|
| Natural language processing | B | 1502 |
| Machine learning | C | 1031 |
| Text mining | B | 1005 |
| Deep learning | C | 401 |
| NLP | B | 294 |
| Sentiment analysis | B | 206 |
| Artificial intelligence | C | 167 |
| Language model | B | 165 |
| Information extraction | C | 128 |
| Text analysis | B | 125 |
| Text classification | B | 124 |
| Classification | C | 119 |
| Social media | C | 115 |
| Data mining | C | 109 |
| Natural language | B | 102 |
| Learning | A | 99 |
| Text analytics | B | 95 |
| Big data | C | 89 |
| Neural networks | C | 85 |
| Information retrieval | C | 78 |
| Electronic health records | NA | 77 |
| Speech recognition | C | 76 |
| Transfer learning | C | 74 |
| Natural language processing (nlp) | B | 73 |
| Topic modelling | B | 73 |
| Processing | C | 71 |
| Ontology | B | 68 |
| BERT | B | 67 |
| Twitter | C | 66 |
| Computational linguistics | B | 65 |
| COVID-19 | NA | 64 |
| Natural | C | 62 |
| Language models | B | 60 |
| Language processing | B | 60 |
| Word embeddings | B | 60 |
| Language modelling | B | 58 |
| Named entity recognition | B | 58 |
| Clustering | C | 57 |
| Text | C | 57 |
| LSTM | B | 53 |
| Neural network | C | 52 |

Using the pool of related author keywords and guided by the aforementioned new search strategies, we next performed searches on Scopus that led us to a set of 9666 papers. Motivated by the richness of the publications returned by Scopus and guided by the findings of [13], we chose not to repeat this comprehensive search on Web of Science or Dimension or any other citation databases. Next, the abstract and title of the publications were manually examined to guarantee that the papers suit the scope of this study. Papers with a focus on analysis of literature review using text mining, conference proceedings, proceeding trend analysis, journal trend analysis, bibliometric analysis papers (systematic reviews), theses, papers with a focus on new text mining or natural language processing

techniques in non-educational settings, and studies that examine the application of text mining and natural language processing in a broad sense were removed. In the end, a total number of 981 publications were selected and used for analysis in this study (the final search term used for this study as well as the resulting BibTex files are available for download at https://zenodo.org/record/5890421#.Yeu92f5BxjE). It's worth to mention that the number of the accepted papers when the final search term is used (981) is considerably larger than the number of the accepted papers (n = 321) when first search terms were used.

The quantitative analysis in this review employs Bibliometric analysis of the selected papers to generate various quantitative results and identify the main research themes. Authors of [14] provide a summary of some of the widely used tools for bibliometric analysis. We used the Bibliometrix R package [15] to conduct the bibliometric analysis for this paper. The package provides various functions for a comprehensive analysis of the selected literature.

## 3. Results

### 3.1. Descriptive Analysis

As mentioned in previous section, the time-span used for this study covers all publications of the 21st century, i.e., 2000–2021. Table 2 shows the number of publication per year. Among the selected studies, there are 377 articles, 18 book chapters, 584 conference papers, and 1 data paper and 1 short survey paper. These papers were authored by a total number of 2745 authors with a ratio of 0.35 papers per author and 2.8 authors per document. The total number of keywords associated to these papers are 6185 with 3960 and 2225 keywords identified as Keyword Plus and author keywords respectively which shows the high topic diversity of the investigated papers (see Table 3 for the top 10 keywords). Table 4 overviews the name and number of papers of top 10 relevant sources where these studies are published. The top ten countries with highest number of publications include USA (391), China (221), India (122), Japan (87), Indonesia (75), United Kingdom (69), Spain (42), Germany (41), Australia (38), and Italy (36). The top 10 countries with highest number of citations are USA (2020), Spain (440), United Kingdom (423), Tunisia (209), Hong Kong (155), China (141), Denmark (114), Netherlands (109), India (104) and Japan (102). See Figure 2 for the affiliation associated to different countries and their areas of focus identified using the keywords used.

**Table 2.** Publication per year.

| Year | Number of Publications |
|---|---|
| 2000–2004 | 18 |
| 2005–2019 | 67 |
| 2010–2014 | 136 |
| 2015–2019 | 346 |
| 2020–January 2022 | 414 |

**Table 3.** List of top 10 keywords and the frequencies.

| Author Keywords | Articles | Keywords-Plus (ID) | Articles |
|---|---|---|---|
| Natural language processing | 140 | Students | 306 |
| Sentiment analysis | 131 | Natural language processing systems | 223 |
| Machine learning | 122 | Data mining | 195 |
| Text mining | 122 | Learning systems | 171 |
| Deep learning | 64 | Sentiment analysis | 160 |
| Artificial intelligence | 38 | Natural language processing | 152 |
| E-learning | 37 | E-learning | 123 |
| Educational data mining | 32 | Teaching | 110 |
| Data mining | 29 | Text mining | 102 |
| Text classification | 28 | Education | 94 |

**Table 4.** Most relevant sources.

| Name | Number of Publications |
|---|---|
| Lecture Notes in Computer Science | 57 |
| ACM International Conference Proceedings Series | 29 |
| Advances in Intelligent Systems and Computing | 24 |
| CEUR Workshop Proceedings | 19 |
| Communication in Computer and Information Science | 17 |
| Journal of Physics: Conference Series | 17 |
| Pervasive Health: Pervasive Computing Technologies for Healthcare | 17 |
| International Journal of Advanced Computer Science and Applications | 11 |
| International Journal of Artificial Intelligence in Education | 11 |
| IEEE Access | 10 |

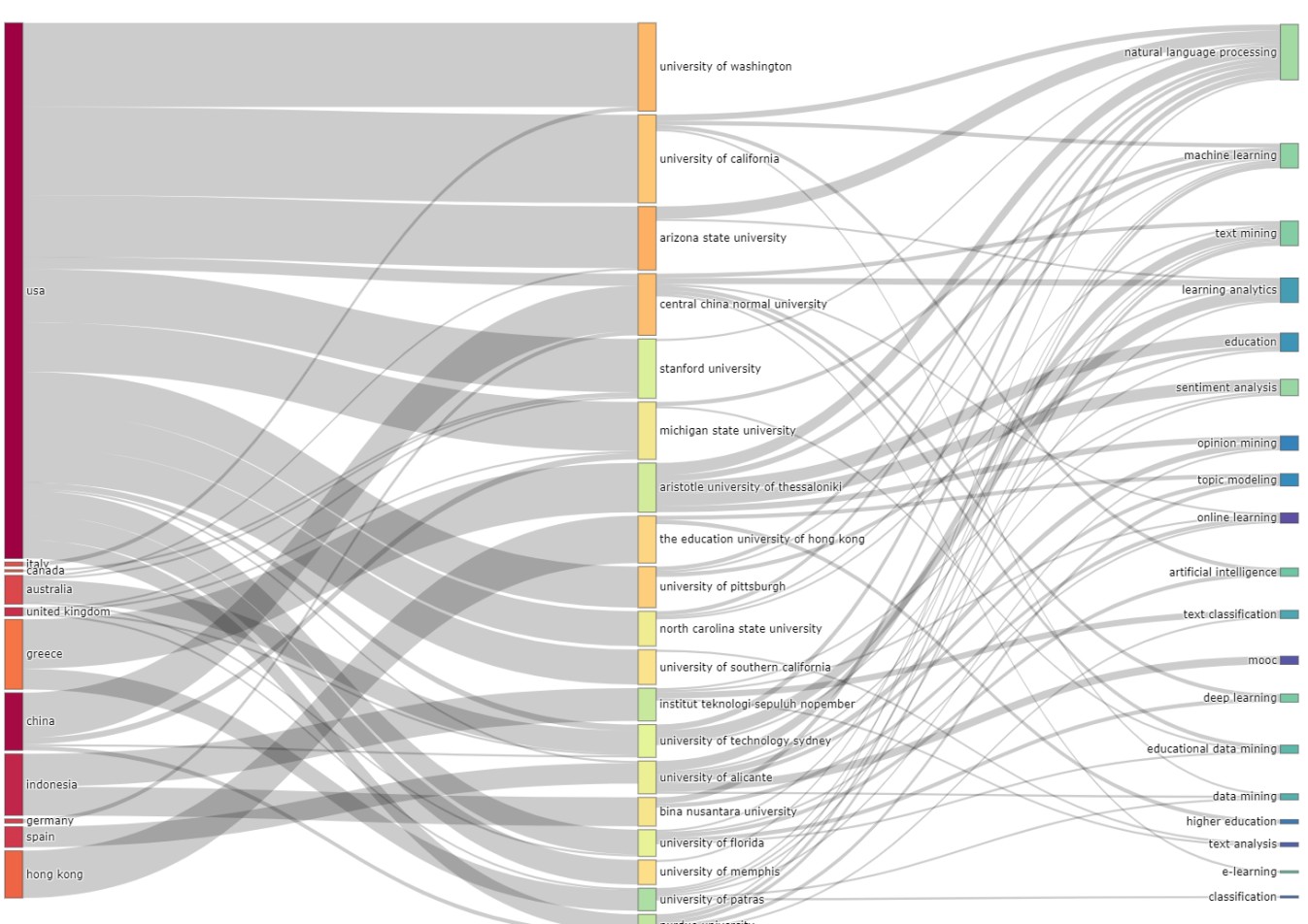

**Figure 2.** The focus area of different educational institutes and their corresponding country.

*3.2. Source Analysis*

Figure 3 shows the top 20 publication venues that form the source of referencing for the papers explored in this study. As can be seen, Computers and Education journal is the number one source of referencing. This journal aims to increase knowledge and understanding of ways in which digital technology can enhance education, through the publication of high-quality research, which extends theory and practice. Another highly

cited source is the International Journal of Artificial Intelligence in Education (IJAIED) which publishes papers concerned with the application of artificial intelligence to education. It aims to help the development of principles for the design of computer-based learning systems. It's interesting how these venues have attracted education researcher's attention over the past 20 years (Figure 4). The journals seem to have gained significant and increasing popularity since 2010 amongst the research community.

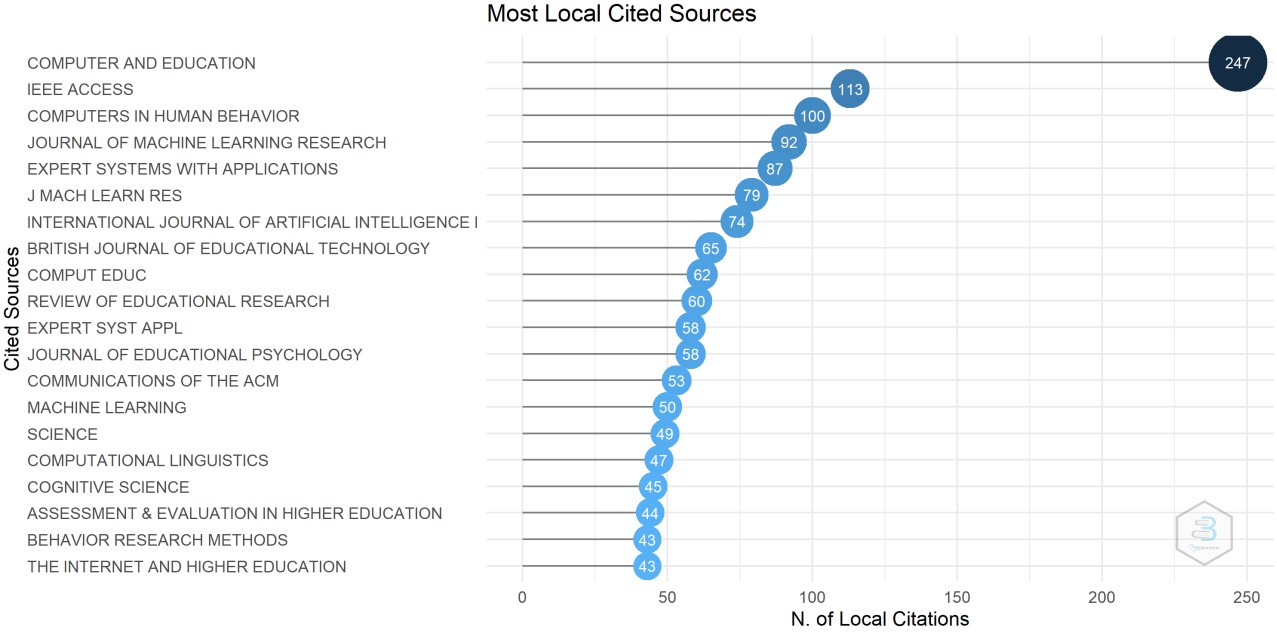

**Figure 3.** Most Local Cited Sources.

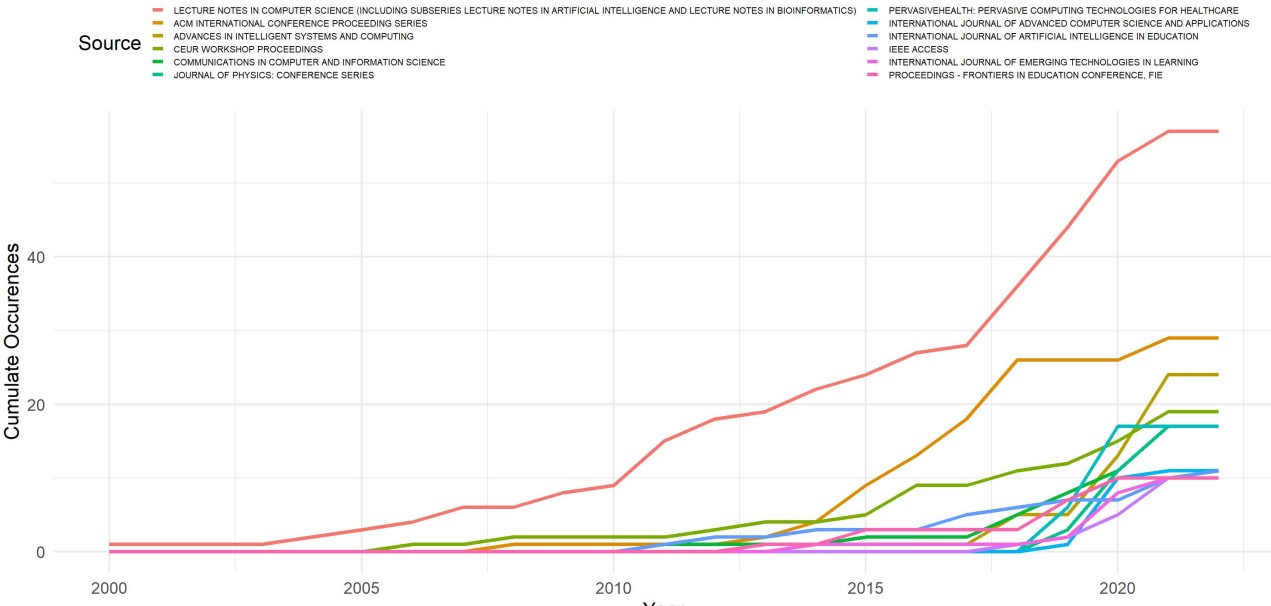

**Figure 4.** Source Dynamics.

### 3.3. Author Analysis

This section identifies top authors in our collection of publications and looks at their annual production. Figure 5 depicts the contribution of top 20 authors to the research behind the application of text mining and natural language processing to learning analytics and educational data mining. Note that this figure does not represent a scoring ladder rather a simple presentation of the name of the authors which have been actively publishing relevant articles hence can provide a good picture of their research output when measured as the publication count. Figure 6 highlights that these authors have been active in the last two decade, i.e., from 2004 to 2022. This result further supports that the domains of text mining and natural language processing have become popular in the last two decades.

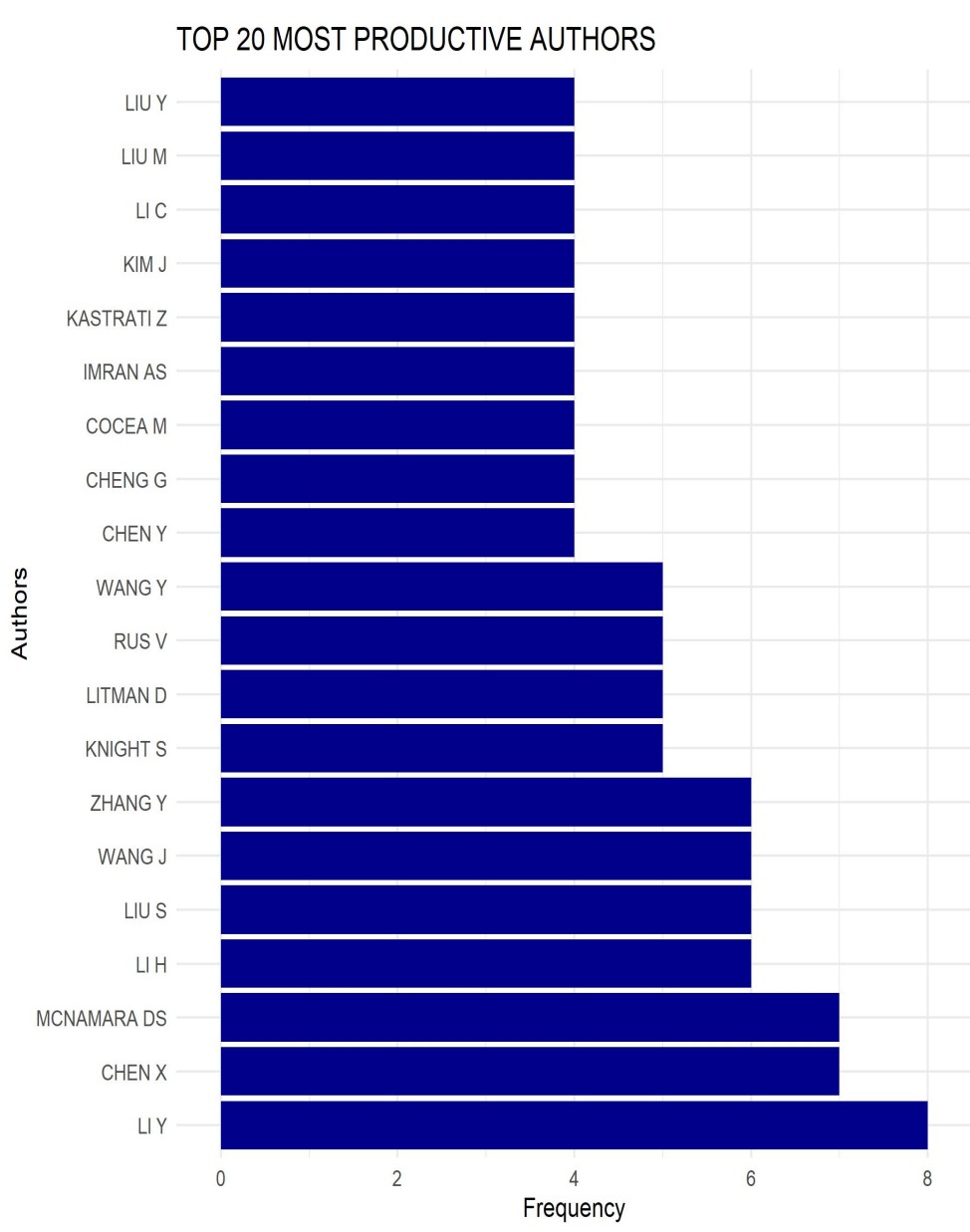

**Figure 5.** Top 20 productive authors based on the number of publications.

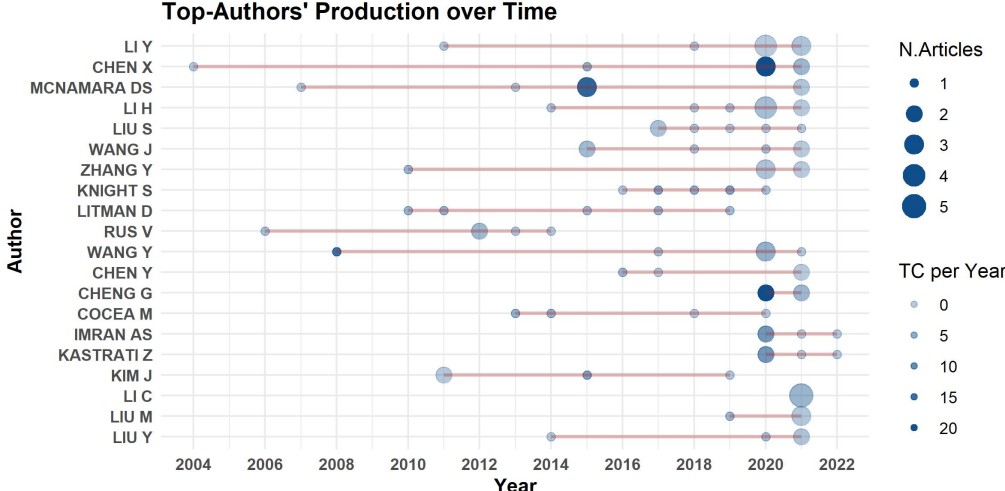

**Figure 6.** Top 20 Authors' Production Over Time.

*3.4. Document Analysis*

The analysis in the previous sections have focused on descriptive analysis of the dataset. This section focuses on the analysis of the research papers on a document level to start addressing our main research questions. Although we analyse all the papers in the final dataset, Table 5 explores the list of 10 highly cited papers that we chose to analyse on a document level in this section to gather their insights into their research themes, results etc. In [16], author examining students' online interaction in a live video streaming environment using data mining and text mining and found the discrepancies as well as similarities in the students' patterns and themes of participation between online questions and online chat messages. Hung [17] investigated the longitudinal trends of academic articles in Mobile Learning (ML) using text mining techniques. McNamara [18] assesses the potential for computational indices to predict human ratings of essay quality. Ref. [19] combined click-stream data and NLP approaches to examine if students' on-line activity and the language they produce in the online discussion forum is predictive of successful class completion. Ref. [20] used natural language processing techniques to evaluate whether text analysis of open responses questions about motivation and utility value can offer additional capacity to predict persistence and completion over and above information obtained from fixed-response items. Ref. [21] synthesised the current methodological approaches to researching collaborative writing and discuss how new text mining tools can enhance research capacity. Ref. [22] aimed to automatically construct the cross references of lecture videos and textual documents so as to facilitate the synchronised browsing and presentation of multimedia information. Ref. [23] presented a new conceptual framework for reflective writing and a computational approach to modelling reflective writing, deriving analytics, and providing feedback. That study also discussed the pedagogical and user experience rationale for platform design decisions and introduced a pilot in a student learning context, with pre-liminary data on educator and student acceptance. Ref. [24] reported on the progress in designing a writing analytics application, detailing the methodology by which informally expressed rubrics are modelled as formal rhetorical patterns, a capability delivered by a novel web application. Ref. [25] used natural language processing tools to build models of students' comprehension ability from the linguistic properties of their self-explanations.

**Table 5.** List of 10 highly cited papers and their citation metrics.

| First Author and Year | Digital Object Identifier | Total Citation (TC) | TC per Year | Ref |
|---|---|---|---|---|
| He W., 2013 | 10.1016/j.chb.2012.07.020 | 120 | 13 | [16] |
| Hung J.L., 2012 | 10.1007/s12528-011-9044-9 | 106 | 11 | [17] |
| Mcnamara D., 2013 | 10.3758/s13428-012-0258-1 | 70 | 8 | [18] |
| Crossley D., 2016 | 10.1145/2883851.2883931 | 61 | 10 | [19] |
| Robinson C., 2016 | 10.1145/2883851.2883932 | 42 | 7 | [20] |
| Yim S., 2017 | 10125/44599 | 37 | 7.4 | [21] |
| Wang F., 2008 | 10.1016/j.patcog.2008.03.024 | 31 | 2 | [22] |
| Gibson A., 2017 | 10.1145/3027385.3027436 | 27 | 5 | [23] |
| BuckinghamShum S., 2016 | 10.1145/2883851.2883955 | 23 | 2 | [24] |
| Allen L., 2015 | 10.1145/2723576.2723617 | 23 | 3 | [25] |

Table 6 provides an overview of the top 50 keywords associated with the set of papers analysed in this systematic review paper. As expected, *natural language processing* is the most frequent word found in the author's keywords. *Sentiment analysis*, *machine learning* and *text mining* form the next group of most frequent author keywords with a occurrence frequency of 131, 122 and 122 respectively. *Deep learning*, *artificial intelligence* and *e-learning* are also among the most repeated keyword with a occurrence frequency of 64, 38 and 37 respectively. The table also shows that more recent methodologies like *Ontology*, *Named Entity Recognition* are among the top-50 keywords and hence gaining popularity. Interestingly, the number of times these keywords have appeared in authors' keywords throughout time have been overall increasing (see Figure 7).

**Table 6.** Top 50 author keywords.

| Rank | Keyword | Frequency | Rank | Keyword | Frequency |
|---|---|---|---|---|---|
| 1 | Natural language processing | 140 | 26 | Neural network | 12 |
| 2 | Sentiment analysis | 131 | 27 | Student feedback | 12 |
| 3 | Machine learning | 122 | 28 | Automated essay scoring | 11 |
| 4 | Text mining | 122 | 29 | Feedback | 11 |
| 5 | Deep learning | 64 | 30 | Intelligent tutoring systems | 11 |
| 6 | Artificial intelligence | 38 | 31 | LSTM | 11 |
| 7 | E-learning | 37 | 32 | Natural language processing (NLP) | 11 |
| 8 | Educational data mining | 32 | 33 | Support vector machine | 11 |
| 9 | Data mining | 29 | 34 | BERT | 10 |
| 10 | Text classification | 28 | 35 | Feature selection | 9 |
| 11 | Learning analytics | 27 | 36 | Natural language | 9 |
| 12 | Education | 26 | 37 | Teaching evaluation | 9 |
| 13 | Topic modelling | 24 | 38 | Text analytics | 9 |
| 14 | Opinion mining | 23 | 39 | Word embedding | 9 |
| 15 | Higher education | 19 | 40 | Word2vec | 9 |
| 16 | Classification | 18 | 41 | Assessment | 8 |
| 17 | NLP | 18 | 42 | Big data | 8 |
| 18 | Text analysis | 18 | 43 | COVID-19 | 8 |
| 19 | MOOC | 16 | 44 | IDA | 8 |
| 20 | Online learning | 15 | 45 | Plagiarism detection | 8 |
| 21 | Chatbot | 13 | 46 | SVM | 8 |
| 22 | Latent dirichlet allocation | 13 | 47 | Twitter | 8 |
| 23 | Learning | 13 | 48 | Named entity recognition | 7 |
| 24 | MOOCs | 13 | 49 | Natural language understanding | 7 |
| 25 | Information retrieval | 12 | 50 | Ontology | 7 |

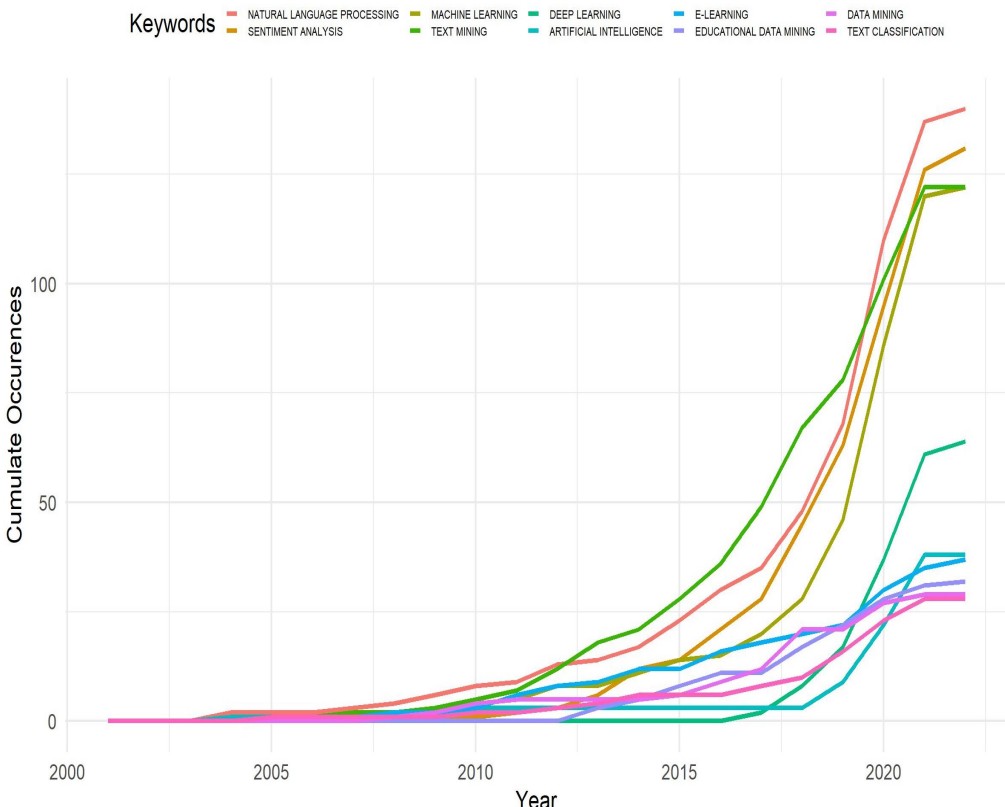

**Figure 7.** Word Dynamics (Top-10 Keywords).

*3.5. Conceptual Structure Analysis*

One of the main objectives of this systematic review is to identify the main themes and topics of interest from the previous studies. A thematic analysis based on co-word network analysis and clustering [26] is performed to identify various research topics in two dimensions of centrality and density. The centrality measures the degree of interaction of a network with other networks. This can be interpreted as a measure of the importance of a theme in the development of the entire research field analysed. The density measures the internal strength of the network and identifies the degree of development of a theme. The analysis quantifies the extant and within ties of keywords with various themes in the dataset [27]. Analysing the keywords from the papers in our dataset using the thematic analysis reveals various topics as per their stage of development and relevance. Figure 8 presents these themes in four quadrants, namely *motor themes*, *Niche themes*, *emerging or declining themes*, and *basic and transversal themes* according to their centrality and density rank. The size of each cluster is determined by the number of times the keywords occurred.

The upper right quadrant presents the Motor themes; well developed themes that are key to the structure of the research field. As can be seen, Text Mining, Educational Data Mining, and Data Mining are the well developed themes that have been used for the analysis of a variety of different types of text data. This is not surprising as the search terms used in collecting the publications highly correlate with these themes.

The upper left quadrant identifies the Niche themes. These are specialised yet marginal themes with respect to the other themes observed in the entire population of the papers investigated. According to the Niche themes quadrant in Figure 8, Language Processing, Language Learning and Automated Grading are identified as specialised themes. These are among those analytical approaches that are well established and yet are slightly marginal to the dominant fields observed in the Motor themes. This indicates that while these techniques are well established they are applied in specialised research cases.

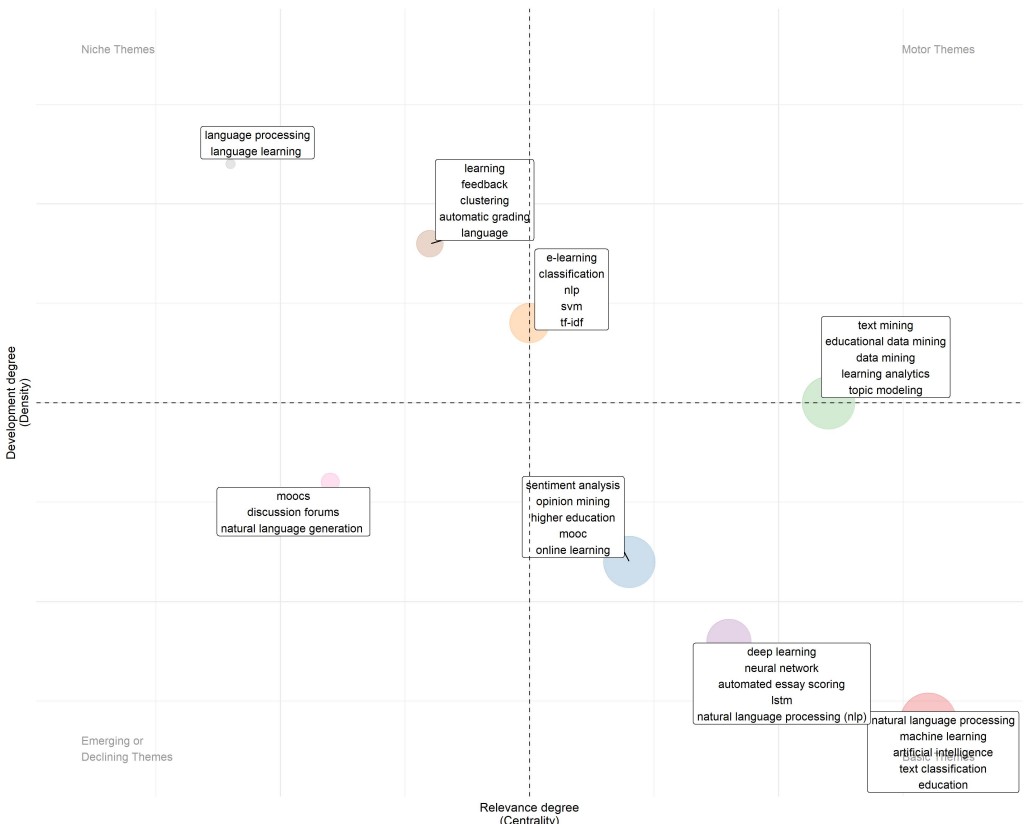

**Figure 8.** Thematic Map.

The lower left quadrant identifies emerging or declining themes which represent the topics which are at the periphery of the research field. Interestingly the analysis identifies textual analysis of MOOCs, Discussion Forums and Natural Language Generation as emerging themes implying that it has the potential to become one of the main themes in learning analytics. This can be attributed to the growing use of online LMS platforms to teach and conduct engagement activities.

Finally the lower right quadrant represents Basic and transversal themes. These themes are regarded as important for the field and are frequently researched. According to the Basic themes, applications of Deep Learning, Neural Network, Natural Language Processing, Machine Learning and Artificial Intelligence in e-learning and education research seems to be essential for learning analytics and educational data mining communities. More specifically, Sentiment Analysis and Opinion Mining of the data collected by higher education institutes seem to be among frequently used analytics techniques. Text Analysis of students' feedback, evaluation of teaching, and higher education research are among those themes which seems to very important in the community but are yet to be developed further and positioned properly in the learning analytics community.

Overall, we can conclude that while research in areas such as Machine Learning, Artificial Intelligence, and Educational Data Mining are regarded as well established research areas, the application of the variety of methods and techniques available in these disciplines are not fully utilised by education research community. Particularly, the niche themes around contextual text analysis are niche and require further attention. This is also evident from the findings present in Figure 9 which plots the top trending topics with keywords appearing at least five times in the dataset. As can be seen, most research trends in the first 10 years of the 21st century are around Writing Analytics using NLP and Text Analytics techniques such as embedding for genre analysis and language learning. It's only recently where the Machine Learning is identified as a research trend. Interestingly, different text analytics techniques such as Topic Modelling and Sentiment Analysis, Writing Analytics, and predictive analytics started to become popular among learning analytics

community. The general field of Computational Linguistics has gained popularity in the last decade with Deep Learning, Information Retrieval and the use of ontologies are among those trends which have just recently gained popularity.

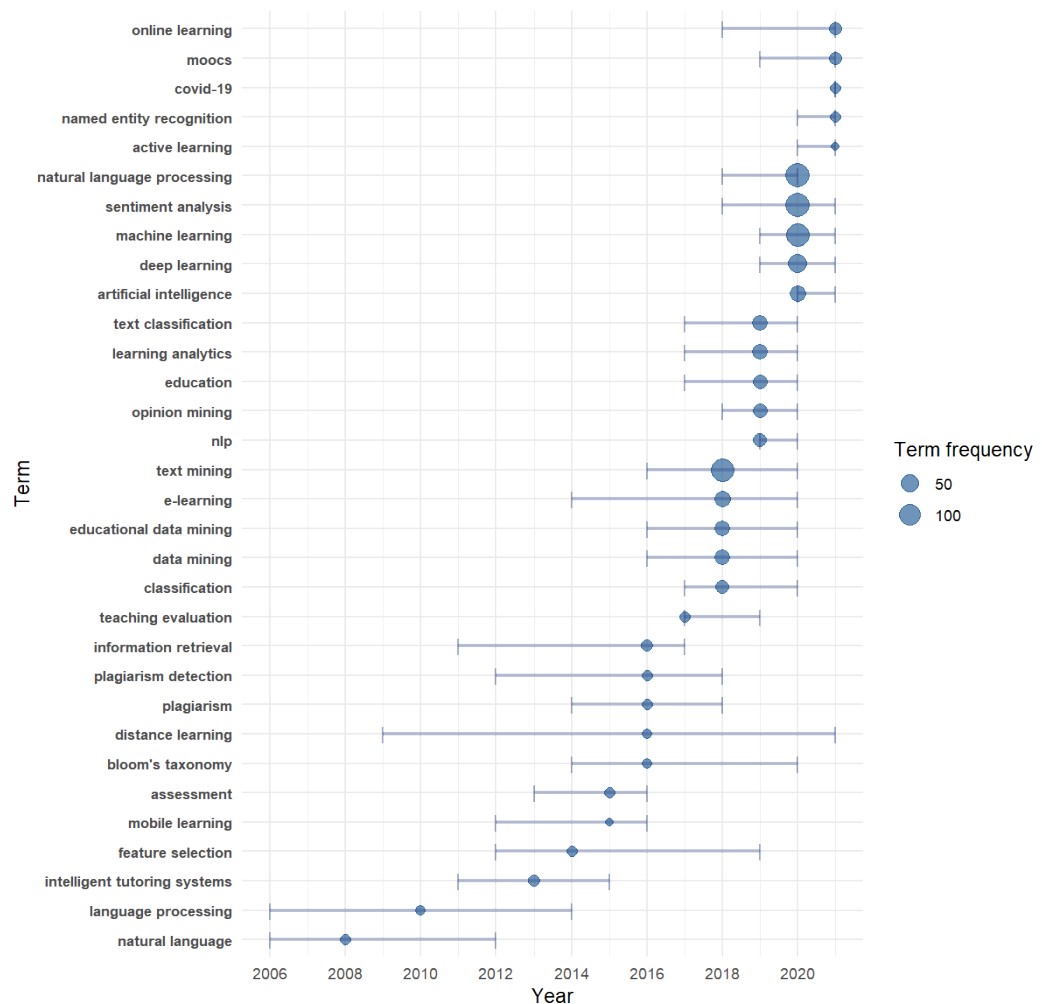

**Figure 9.** Top trending topics within collection of publications (2000–2020).

To provide a better picture of the relationship between different topics, we performed unsupervised machine learning based visualisation of the author keywords (see Figure 10). This visualisation represents a clustering of the top 50 author keywords where different author keywords are grouped using **M**ultiple **C**orrespondence **A**nalysis (MCA) method resulting in a conceptual structure map of the publications investigated in this study. The algorithm generates three clusters, the first and the main group of publications (red cluster) are more focused around the application of text mining and natural language processing in the analysis of survey data, curricula, and the student data collected from e-learning environments. The second group of papers (the blue cluster), which includes a small proportion of the publications in our dataset are those publications that are concerned with applications of topic modelling techniques in educational context. The third cluster (the green cluster) identifies the various use cases which are assessed using text mining methods, it also relates them with the use of social media.

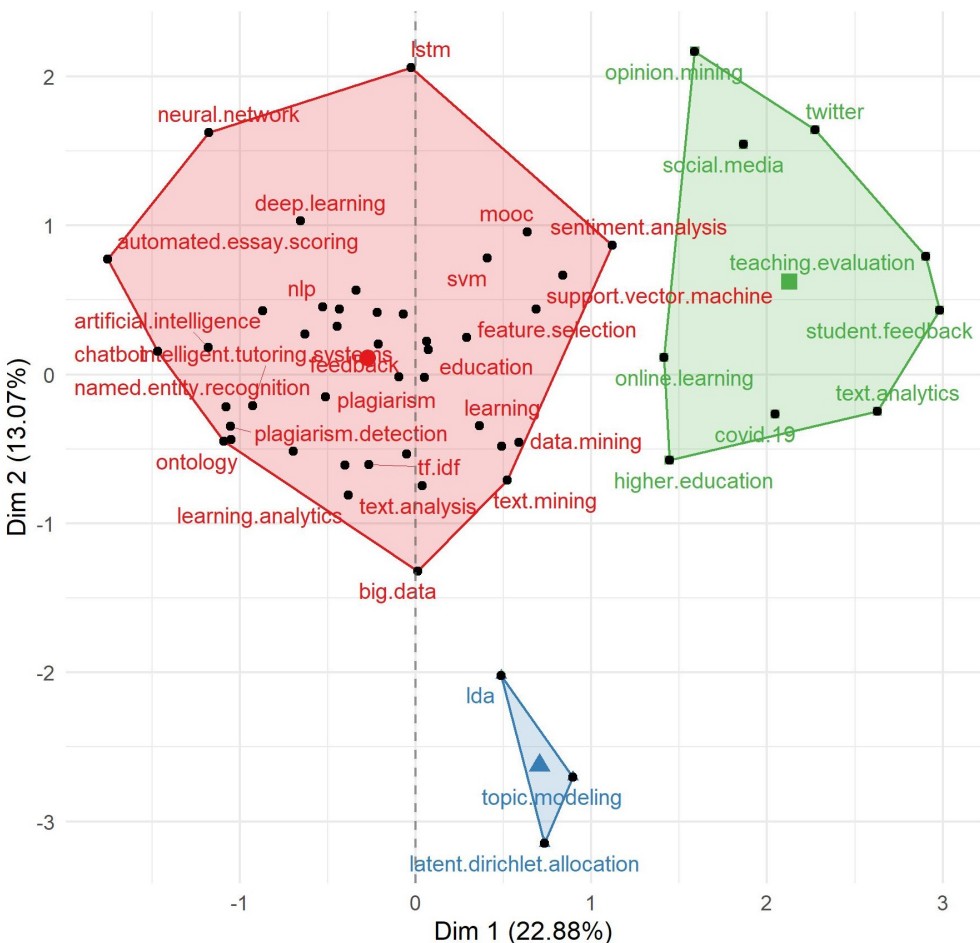

**Figure 10.** Conceptual structure map generated by MCA method.

## 4. Discussion and Conclusions

This study aimed to systematically review peer reviewed research papers published in 21st century that use text mining or natural language processing in education research. Guided by PRISMA protocol, we analyse the metadata of a collection of 981 publications using Bibliometrics software and report on the different aspects of the use of natural language processing and text mining in different aspects of education research. We report on the scientific contribution of different countries and higher education institutes to the field. Our extensive analysis on the conceptual structure and themes explored by the publications investigated in this study provides a high level view of the topics that have been of interest to the education research community. This in turn provides an understanding of the themes that have attracted less attention for the education research community as well as the degree of relevance of these themes. More specifically, the cluster analysis of the publications highlights what different techniques and areas of applications are interconnected which can be used as a guide to identify research gaps. Lastly, the Bibliometrix software used for the systematic review of the publications was found to be a useful tool that can enable simple and reliable bibliometric analysis.

While the systematic reviews enable unravelling useful information about different aspects of the research associated to educational text mining, the study is by nature limited to a certain number of caveats. Similar to other data-driven data analysis approaches, incomplete or inaccurate data can result in incorrect or misleading conclusions. While an exhaustive publication search process was used to find the papers, it could have been the case that some of the publications that are in fact highly influential have not been identified. Another reason for possible omission of the related research papers in the publication search process could be due to the fact that some of the key contributing publication venues

might not be indexed by abstract and citation databases (Scopus and Web of Science). The incompleteness of the data can also occur on a metadata level where some of the information associated to the data fields are not present for some of the publications. The exclusion of the grey literature, lack of appropriate critical appraisal of included study validity and inappropriate synthesis are among other issues that can naturally impact the quality of findings. The use of PRISMA as a high-quality guidance, careful design of the research strategy, and careful examination of the literature for identification of the grey literature papers are among few steps that were used in this study to guaranty a high quality of the data, methods used and consequently the findings presented in this paper.

Our findings show that while a certain number of text mining techniques have been applied to address different research questions related to teaching and learning, there still is a need for more replication studies to explore the results reported in these papers in different contexts. Based on the results of the thematic analysis, it is evident that there are certain areas that have been given less attention by the research community hence a more developed stage for the research associated to these areas is yet subject to future research efforts.

**Author Contributions:** Conceptualization, A.A., A.S., M.B., and M.G.; methodology, A.A. and A.S.; software, A.A. and A.S.; validation, A.A. and A.S.; formal analysis, A.A. and A.S.; investigation, A.A.; resources, A.S., M.B., and M.G.; data curation, A.A.; writing—original draft preparation, A.A.; writing—review and editing, A.A., A.S., M.B., and M.G.; visualization, A.A. and A.S.; supervision, A.S. and M.B.; project administration, A.S. and M.B. All authors have read and agreed to the published version of the manuscript.

**Funding:** This project was funded by an Australian SIEF STEM+ business fellowship in conjunction with the industry partner Cinglevue International Pty Ltd.

**Institutional Review Board Statement:** Not applicable.

**Data Availability Statement:** The data presented in this study are openly available in Zenodo at 10.5281/zenodo.5890421.

**Conflicts of Interest:** The authors declare no conflict of interest.

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
