# Peer review of "Text Mining in Education—A Bibliometrics-Based Systematic Review"

_education, doi:10.3390/educsci12030210_

Round 1

Reviewer 1 Report

Article is a standard bibliometric review. Only, review the text because of some words appear together.

Author Response

We thank the reviewer for their feedback as the comment has helped us improve the quality of our paper.

Regarding the English language use, we have checked the paper again thoroughly to make sure that there are no typos and grammatical mistakes are present in the article.

Reviewer 2 Report

This manuscript provides a systematic review of NLP and text analytics in relation to education since 2000.  The authors have described the topics and themes of research in this area, and their relations.  The paper is well written and the topic is timely, with potential to provide valuable insights to researchers who are interested in automated text analytics. 

My primary concern regards the search terms.  My sense is that many papers may have been missed because the search terms were limited. Given that there were not any false alarms identified (articles mistakenly identified), that implies that many must have been missed.  One approach would have been to use the initial list to augment the search terms (e.g., using the common terms from Figure 10).  Some terms that I noticed missing were: 

computational linguistics language model writing analytics Automated writing evaluation Automated essay scoring NLP comprehension computational + text   I noticed several papers missing due to not using these search terms. On the one hand, a systematic review is what it is - the authors used those terms and that's what comes out. On the other hand, it might be worth the while to expand the search to ensure a comprehensive picture of the field. I'd leave it to the editor to decide between requesting an expanded search or simply noting the limitations and potential missed papers.    Regarding the manuscript as it stands: The document analysis could be more insightful - rather than listing what the articles do, it would be more informative to group the articles - e.g., I would assess the semantic connections between the papers to cluster them visually.  From that section, I don't have a sense of the themes that are emerging, but rather just a list of topics.    In addition, it is not clear how Figure 9 was generated - what analysis was used?  This figures also needs to be more thoroughly explained and discussed.    Perhaps most important - the authors need to make these data, and the list of papers available in open source; i.e., the data would have to be publicly available.    

Author Response

We thank the reviewer for their comments as the comments helped us improve the quality of our paper.

Regarding the English language use, we have checked the paper again thoroughly to make sure that there are no typos and grammatical mistakes are present in the article. Here we brief what is our response to each comment, and what action has been made to the paper to reflect on the given comment:

Regarding the search terms, we took the reviewer’s comment as an advice (thank you for that) and constructed a list of keywords based on most frequent terms represented in Figure 10. This led to the identification of a list of papers that were not previously returned by our initial search. These papers are now included in the analysis and are present in the appendix of the paper and the results section and other parts of the paper are accordingly updated as well to reflect on the new findings. We believe that the extensive examination of a large number of paper through the updated search terms has given us a much wider lens over the state of the art of the field.

With respect to the suggestion regarding grouping the articles, we have taken reviewer’s comment as a guide and added a new figure that explores the semantic relationship of the papers in a visual fashion.

We have also explained how figure 9 is generated and added to the discussion related to the figure in the text.

Finally, we have the title of the papers used in our analysis as well as the bibTex file containing citation information of these papers on a public website (available in the paper as a link for free download). Once again, we thank the reviewer for their valuable comments.

Reviewer 3 Report

Review of ‘Title Text Mining in Education - A Systematic Review’

  1. The title and abstract are correct although the abstract is missing some important features like results, discussion, and conclusion (e.g., include lines 63-68 in the abstract). Also, something more could be written about methods (systematic review) in the abstract. Keywords are fine. Please consider the use of bibliometrics in the paper’s title.
  2. Regarding Introduction, the section is correct, including the research questions that guided the research. I suggest that ‘learning analytics’ concept should be presented in more detail, as it will be an essential concept in the paper (see 276-278 to understand this observation).
  3. In Methodology section, the PRISMA guideline should be updated with its most recent version (see Page, M. J., McKenzie, J. E., Bossuyt, P. M., Boutron, I., Hoffmann, T. C., Mulrow, C. D., Shamseer, L., Tetzlaff, J. M., Akl, E. A., Brennan, S. E., Chou, R., Glanville, J., Grimshaw, J. M., Hróbjartsson, A., Lalu, M. M., Li, T., Loder, E. W., Mayo-Wilson, E., McDonald, S., … Moher, D. (2021). The PRISMA 2020 statement: An updated guideline for reporting systematic reviews. BMJ, 372, n71. https://doi.org/10.1136/bmj.n71). Also, the choice of Web of Science and SCOPUS must be explained. Search expressions could have been copied from the databases to improve reproducibility (Lines 98-101). Figure 1 caption should include the PRISMA citation (namely, the 2021 update). In Figure 1 is also mentioned a set of 10 publications for qualitative analysis, but the rationale of this option in not included in Methodology. Please explain it.
  4. Regarding Results, please correct lines 182-183 reference to the Figure’s numbering. Line 199 change explores for depicts. Line 225 change and overview for an overview. Figure 7 is very difficult to read. Please find another visualization strategy. In general, it is a very good section.
  5. In Discussion and conclusion section, please see Line 331 (data-deriven?!?). The paper does not have a true discussion of results, as they are not compared with the previous literature. I suggest that some of the results might be compared with the references cited in the Introduction section.

Author Response

We thank the reviewer for their comments as the comments helped us improve the quality of our paper.

Regarding the language use, we have checked the paper again thoroughly to make sure that there are no typos and grammatical mistakes are present in the article. One of our authors whose first language is English has also read the paper and made appropriate changes to the language.

Regarding the abstract, we have updated the abstract of the paper to include the suggested point in it. We have also updated the title of the paper by using the word “bibliometrics” in the title.

We also have explored the concept of “learning analytics” in the paper in more details (introduction section of the paper).

We have updated the PRISMA guideline’s reference to the reference suggested by the reviewer. Also, we now have explained why we have chosen Web of Science as our choice of publication database. We have placed used search terms in the manuscript as well. We have added the PRISMA citation in the caption of Figure 1 as per advised. We also have explained why we have choses the publication count of 10 in the paper as per suggested by the reviewer.

In the results section, we have corrected the reference to figure numbering (lines 182 -183). Also, we have corrected the other grammatical mistakes detected by the reviewer. Also, we have replaced Figure 7 with a table to make it easier to make it easier to read.

Lastly, we have updated the result section based on the comment provided by the reviewer. Once again, we thank the reviewer for their valuable comments.

This manuscript is a resubmission of an earlier submission. The following is a list of the peer review reports and author responses from that submission.